# Long non-coding RNA *GRASLND* enhances chondrogenesis via suppression of the interferon type II signaling pathway

Nguyen PT Huynh[1,2,3,4], Catherine C Gloss[1,2,4], Jeremiah Lorentz[1,2,4], Ruhang Tang[1,2,4], Jonathan M Brunger[5], Audrey McAlinden[1,2,4], Bo Zhang[4], Farshid Guilak[1,2,4]*

[1]Department of Orthopaedic Surgery, Washington University, St Louis, United States; [2]Shriners Hospitals for Children, St. Louis, United States; [3]Department of Cell Biology, Duke University, Durham, United States; [4]Center of Regenerative Medicine, Washington University, St Louis, United States; [5]Department of Biomedical Engineering, Vanderbilt University, Nashville, United States

**Abstract** The roles of long noncoding RNAs (lncRNAs) in musculoskeletal development, disease, and regeneration remain poorly understood. Here, we identified the novel lncRNA *GRASLND* (originally named *RNF144A-AS1*) as a regulator of mesenchymal stem cell (MSC) chondrogenesis. *GRASLND*, a primate-specific lncRNA, is upregulated during MSC chondrogenesis and appears to act directly downstream of SOX9, but not TGF-β3. We showed that the silencing of *GRASLND* resulted in lower accumulation of cartilage-like extracellular matrix in a pellet assay, while *GRASLND* overexpression – either via transgene ectopic expression or by endogenous activation via CRISPR-dCas9-VP64 – significantly enhanced cartilage matrix production. *GRASLND* acts to inhibit IFN-γ by binding to EIF2AK2, and we further demonstrated that *GRASLND* exhibits a protective effect in engineered cartilage against interferon type II. Our results indicate an important role of *GRASLND* in regulating stem cell chondrogenesis, as well as its therapeutic potential in the treatment of cartilage-related diseases, such as osteoarthritis.

**\*For correspondence:**
guilak@wustl.edu

**Competing interests:** The authors declare that no competing interests exist.

## Introduction

Articular cartilage is an aneural, avascular tissue and has little or no capacity for intrinsic repair (*Sophia Fox et al., 2009*), and there are currently no effective procedures that result in long-term cartilage restoration. Furthermore, focal cartilage or osteochondral lesions generally progress to osteoarthritis (OA), a progressive degenerative disease characterized by changes in the articular cartilage and remodeling of other joint tissues such as the synovium and subchondral bone. Thus, there remains an important need for regenerative therapies that can enhance cartilage repair through tissue engineering or cell therapy approaches (*Huynh et al., 2018a*; *Glass et al., 2014*; *Brunger et al., 2017a*; *Brunger et al., 2017b*; *Brunger et al., 2014*; *Adkar et al., 2017*; *Bhumiratana et al., 2014*).

In this regard, adult stem cells such as bone marrow-derived mesenchymal stem cells (MSCs) or adipose-derived stem cells (ASCs) provide a readily accessible source of multipotent cells that show significant promise for regenerative medicine (*Gimble and Guilak, 2003*; *Erickson et al., 2002*; *Awad et al., 2004*; *Caplan, 1991*). Under defined culture conditions supplemented with Transforming Growth Factor Beta 3 (TGF-β3), MSCs produce a cartilaginous matrix that is rich in glycosaminoglycans (GAGs) and collagen type II (COL2) (*Mackay et al., 1998*; *Johnstone et al., 1998*). However, the complete pathway involved in MSC chondrogenesis has not been fully deciphered. A

detailed understanding of the gene regulatory networks that control this process could provide new insights that accelerate and improve cartilage regeneration from endogenous stem cells or exogenously implanted MSCs.

Increasing evidence suggests that the gene regulatory pathways involved in stem cell differentiation may rely not only on protein-coding RNAs, but also on non-coding RNAs (ncRNAs). ncRNAs were initially difficult to identify because they did not possess open reading frames and were not evolutionarily highly conserved (*Lander et al., 2001*). In one of the first landmark studies, chromatin-state mapping was used to identify transcriptional units of functional large intervening non-coding RNAs (lincRNAs) that were actively transcribed in regions flanking protein-coding loci (*Guttman et al., 2009*), and follow-up loss-of-function studies indicated that these lincRNAs were indeed crucial for the maintenance of pluripotency in embryonic stem cells (*Guttman et al., 2011*). There is a growing understanding of long non-coding RNA (lncRNA) function in a multitude of tissues and cellular processes. For example, detailed mechanistic studies on the role of lncRNAs in X chromosome inactivation (*Lee and Bartolomei, 2013*) or in nervous system development and functions (*Ng et al., 2012*; *Briggs et al., 2015*) have been reported previously. However, knowledge of their roles in the musculoskeletal system, particularly in chondrogenesis, remains limited. Only a handful of functional studies have been carried out in this regard. For example, lncRNA-HIT (HOXA Transcript Induced by TGFβ) (*Carlson et al., 2015*) has been shown to play a role in epigenetic regulation during early limb development. Other studies have implicated a specific lncRNA, ROCR (Regulator of Chondrogenesis RNA) (*Barter et al., 2017*) in activity upstream of SRY-Box 9 (SOX9) and in the regulation of chondrocyte differentiation (*Huynh et al., 2017*).

As one of their many modes of actions, lncRNAs are also known to regulate and modulate various signaling cascades involved in the control of gene regulatory networks. Therefore, there may exist a connection between lncRNA candidates and signaling pathways previously known to play a role in the development of the musculoskeletal system. More specifically, there is growing evidence for the role of interferon (IFN) in skeletal tissue development and homeostasis (*Dieudonne et al., 2013*; *Rostovskaya et al., 2018*; *Takayanagi et al., 2002a*; *Takayanagi et al., 2002b*; *Li, 2013*; *Sahni et al., 1999*; *Jang and Baik, 2013*; *Xiao et al., 2004*; *Sahni et al., 2001*). There are two main types of IFN. Type I IFN includes mainly IFN alpha (IFN-α) and IFN beta (IFN-β), which form complexes with Interferon Alpha and Beta Receptors (IFNARs), activating the Janus Kinase/Signal Transducers and Activators of Transcription (JAK/STAT) pathway by phosphorylation of STAT1 (Signal Transducer and Activator of Transcription 1) and STAT2 (Signal Transducer and Activator of Transcription 2). Phosphorylated STAT1/STAT2 then form complexes with IRF9 (IFN Regulatory Factor 9) and translocate into the nucleus to activate downstream targets via the interferon-stimulated responsible element (ISRE) DNA-binding motif. Type II IFN, on the other hand, relies on activation of the JAK/STAT pathway following the binding of IFN gamma (IFN-γ) to Interferon Gamma Receptors (IFNGRs). This process subsequently results in the phosphorylation and dimerization of STAT1, which translocates into the nucleus and induces downstream targets via the gamma activated sequence (GAS) DNA-binding element (*Brierley and Fish, 2002*; *Hertzog et al., 1994*; *Hu and Ivashkiv, 2009*). Although IFN are widely known for their antiviral response, they can also act in other aspects of cellular regulation (*Hertzog et al., 1994*). Interestingly, IFN-γ has been implicated in non-viral processes, most notably due to its priming effect in auto-immune diseases such as lupus nephritis, multiple sclerosis, or rheumatoid arthritis (*Green et al., 2017*). An additional goal of this study was to elucidate the link between IFN-γ and our lncRNA candidate, and how this interaction could potentially play a role in MSC chondrogenesis and cartilage tissue engineering.

In a recent publication, we used high-depth RNA sequencing to map the transcriptomic trajectory of MSC chondrogenesis (*Huynh et al., 2018b*). This dataset provides a unique opportunity to identify candidate genes for subsequent functional characterization as regulators of chondrogenesis. Here, we used bioinformatic approaches to integrate our RNA-seq data with other publicly available datasets, applying a rational and systematic data-mining method to define a manageable list of final candidates for follow-up experiments. As a result, we identified *RNF144A-AS1* as a crucial regulator of chondrogenesis and propose the name Glycosaminoglycan Regulatory ASsociated Long NoncoDing RNA (*GRASLND*). We showed that *GRASLND* enhances chondrogenesis by acting to suppress the IFN-γ signaling pathway, and that this effect was prevalent across different adult stem cell types and conditions. Together, these results highlight novel roles of *GRASLND* and its modulation

of IFN in stem cell chondrogenesis, as well as its therapeutic potential to enhance cartilage regeneration.

## Results

### *GRASLND* is crucial to and specifically upregulated in chondrogenesis

First, we utilized our published database on MSC chondrogenesis (GSE109503) (*Huynh et al., 2018b*) to identify lncRNA candidates. We investigated the expression patterns of MSC markers (*ALCAM, ENG, VCAM1*), chondrogenic markers (*ACAN, COL2A1, COMP*), and SOX transcription factors (*SOX5, SOX6, SOX9*) (*Figure 1—figure supplement 1A*). Pearson correlation analysis revealed 141 lncRNAs whose expression was highly correlated to those of MSC markers, 40 lncRNAs to chondrogenic markers, and 17 lncRNAs to SOX transcription factors (*Figure 1—figure supplement 1B,C*). Among those, *LOXL1-AS1* and *MIR4435-1HG* were downregulated and *RP11-366L20.2* and *GRASLND* were upregulated upon ectopic SOX9 overexpression (*Table 1* and *Supplementary file 1*) (GSE69110; *Ohba et al., 2015*). To validate the functions of these lncRNAs in chondrogenesis, we systematically designed small hairpin RNAs (shRNAs) targeting each candidate and assessed the knockdown effect after 21 days of chondrogenic induction. We successfully designed two target shRNAs for *LOXL1-AS1*, *MIR4435-1HG*, and *GRASLND*, and one target shRNA for *RP11-366L20.2* (*Figure 1B*, *Figure 1—figure supplement 2A–C*, *Figure 1—source data 1*). We showed that knockdown of two out of three MSC-related lncRNAs did not influence the production of glycosaminoglycans (GAG), an important extracellular matrix component in cartilage (*Figure 1—figure supplement 2A–C*). Although these lncRNAs may have other regulatory functions in MSCs, their roles in chondrogenesis appeared to be minimal. Moreover, we found that lower levels of MSC-correlated lncRNAs did not prime the MSCs toward chondrogenesis. However, knockdown of *GRASLND* (alias *RNF144A-AS1* [*RNF144A Antisense RNA 1*]) resulted in decreased expression of chondrogenic markers (*COL2A1, ACAN*) and in upregulation of apoptotic (*CASP3*) and cellular senescence (*TP53*) markers (*Figure 1A,B*). This effect was not due to nonspecific cytotoxicity of the examined shRNAs, as released levels of lactase dehydrogenase (LDH) were similar among control and shRNA-expressing cells (*Figure 1—figure supplement 2D*; *Riss et al., 2004*). In addition, biochemical assays indicated a reduction in both GAG deposition (p<0.0001) and DNA and GAG/DNA levels (p<0.001) (*Figure 1C–E*). Histologically, we observed the same phenotypic loss of GAG and collagen type II in the extracellular matrices (ECM) of pellet samples with *GRASLND* targeted shRNAs, while the scrambled controls displayed explicit staining of these proteins (*Figure 1F*). Taken together, these data indicate that *GRASLND* may be required for both cellular proliferation and cartilage-like matrix production.

To establish whether *GRASLND* expression is specific to chondrogenesis or involved in other differentiation pathways, MSCs were induced towards adipogenic, osteogenic, or chondrogenic lineages, and *GRASLND* expression was measured at various timepoints throughout these processes. Successful differentiation was observed with an increase in lineage-specific markers: *PPARG*

**Table 1.** Long non-coding RNA candidates shortlist.

| Gene symbol | Gene name | ENSEMBL gene ID | Relationship to MSC chondrogenesis | Relationship to SOX9 |
|---|---|---|---|---|
| LOXL1-AS1 | LOXL1 antisense RNA 1 | ENSG00000261801 | Correlated with MSC marker expression | Downregulated upon SOX9 overexpression |
| MIR4435-2HG gene synonym: MIR4435-1HG | MIR4435-2 host gene | ENSG00000172965 | Correlated with MSC marker expression | Downregulated upon SOX9 overexpression |
| HMGA2-AS1 gene synonym: RP11-366L20.2 | HMGA2 antisense RNA 1 | ENSG00000197301 | Correlated with MSC marker expression | Upregulated upon SOX9 overexpression |
| RNF144A-AS1 Referred to as *GRASLND* in this manuscript | RNF144A antisense RNA 1 | ENSG00000228203 | Correlated with chondrogenic marker expression | Upregulated upon SOX9 overexpression |

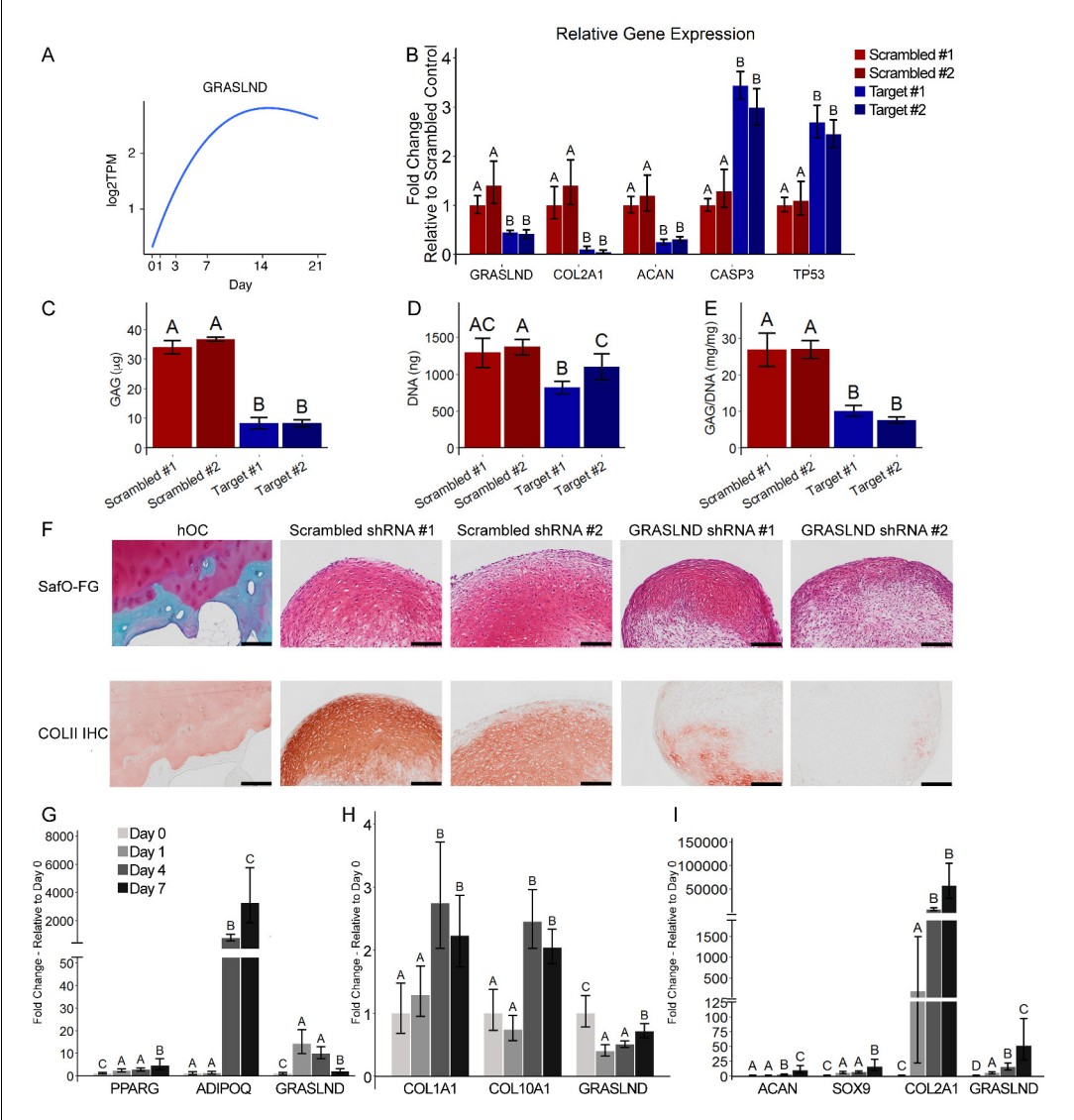

**Figure 1.** *GRASLND* is important and specifically upregulated in MSC chondrogenesis. (**A**) Expression pattern of *GRASLND* in chondrogenesis (GSE109503; *Huynh et al., 2018b*). Log2TPM: log transformed value of transcripts per million (TPM). (**B**) Effect of *GRASLND* knockdown on chondrogenic, apoptotic, and cell-cycle-inhibition markers (n = 5). (**C–E**) Effect of *GRASLND* knockdown on pellet matrix synthesis (n = 5). (**F**) Representative histological images of day 21 MSC pellets. Scale bar = 200 µm. SafO-FG, SafraninO-Fast Green staining; COLII IHC, collagen type II immunohistochemistry; hOC, human osteochondral control. (**G–I**) qRT-PCR analysis of MSC samples cultured in (**G**) the adipogenic condition (n = 6), (**H**) the osteogenic condition (n = 6), and (**I**) the chondrogenic condition (n = 3–4). One-way ANOVA followed by Tukey post-hoc test (α = 0.05). Groups of different letters are statistically different from one another.

The online version of this article includes the following source data and figure supplement(s) for figure 1:

**Source data 1.** shRNA target sequences (5′ – sequence – 3′).
**Source data 2.** qRT-PCR sequencing primers (5′ – sequence – 3′).
**Figure supplement 1.** Identification of lncRNA candidates.
**Figure supplement 2.** Functional validation of identified lncRNA candidates.

(*Peroxisome Proliferator Activated Receptor Gamma*) and *ADIPOQ* (*Adiponectin, C1Q And Collagen Domain Containing*) for adipogenesis, *COL1A1* (*Collagen Type I Alpha 1 Chain*) and *COL10A1* (*Collagen Type X Alpha Chain 1*) for osteogenesis, and *ACAN* (*Aggrecan*), *SOX9* (*SRY-Box 9*) and *COL2A1* (*Collagen Type II Alpha Chain 1*) for chondrogenesis (*Figure 1G–I*). We found that *GRASLND* expression was particularly enriched as chondrogenesis progressed (*Figure 1I*). By contrast, *GRASLND* peaked at earlier timepoints during adipogenesis but decreased at later time points

(*Figure 1G*), and downregulated when MSCs underwent osteogenic induction (*Figure 1H*), indicating that *GRASLND* is specifically upregulated in chondrogenesis. Furthermore, we speculate that *GRASLND* may display inhibitory effects on osteogenesis and adipogenesis, being downregulated during these processes.

To validate these gene expression findings, we performed RNA fluorescence in situ hybridization (FISH) throughout the time course of MSC chondrogenesis. Pellets exhibited *GRASLND* FISH signals at later time points during chondrogenic differentiation, consistent with the RNA-seq data (*Figure 2A*). Next, to confirm the subcellular location of *GRASLND*, we performed qRT-PCR on isolated nuclear and cytoplasmic fractions of day 21 MSC pellets (*Figure 2B*). We compared the subcellular expression patterns of *GRASLND* to those of *NEAT1* (*Nuclear Paraspeckle Assembly Transcript 1*) and *GAPDH* (*Glyceraldehyde 3-Phosphate Dehydrogenase*). *NEAT1* is a lncRNA previously characterized as localizing at the nucleus (*Clemson et al., 2009*; *Sasaki et al., 2009*), and *GAPDH* is an mRNA and thus should be exported to the cytoplasm for protein synthesis. Consistent with previous findings, *NEAT1* displayed lower expression in the cytoplasmic fraction compared to the nuclear fraction, in contrast to *GAPDH*. *GRASLND* exhibited higher expression in the cytoplasm, indicating a cytoplasmic subcellular location. Our finding was recapitulated by RNA in situ hybridization followed by confocal microscopy (*Figure 2C*). Interestingly, as *GRASLND* showed punctate labeling, we speculate that this lncRNA may function in the form of an RNA–protein complex.

## Characterization of *GRASLND*

We examined the characteristics of *GRASLND* by first exploring its evolutionary conservation. Except for exon 1, the genomic region of *GRASLND* (displayed as *RNF144A-AS1* in the UCSC Genome Browser) is highly conserved in primates (*Homo sapiens, Pan troglodytes,* and *Rhesus macaque*) whose common ancestor can be traced back to 25 million years ago (*Gibbs et al., 2007*), while sequences are less conserved in other mammals (*Figure 3A*). This suggests that *GRASLND* may belong to a group of previously identified primate-specific lncRNAs (*Derrien et al., 2012*; *Necsulea et al., 2014*).

Per GENCODE categorization, the AS (antisense) suffix indicates a group of lncRNAs that are positioned on the opposite strand, with overlapping sequences to their juxtaposed protein-coding genes. Often, these lncRNAs play a role in regulating the expression of their protein-coding counterparts (*Huynh et al., 2017*). Therefore, we set out to examine whether this is also the case for *GRASLND* (alias *RNF144A-AS1*) (*Figure 3B–C*). Neither knockdown nor overexpression of *GRASLND* affected RNF144A transcript levels in MSCs cultured with or without TGF-β3. Moreover, RNF144A protein levels also remained unaffected by variations of *GRASLND* levels, as indicated by western blot (*Figure 3D* and *Figure 3—figure supplement 1*). These results indicate that *GRASLND* is not involved in the regulation of RNF144A. For these reasons, we proposed that *GRASLND* should be used to refer to the lncRNA in place of *RNF144A-AS1*.

Next, we explored the signaling axis of *GRASLND*. Data mining and computational analysis on earlier published data suggested that *GRASLND* was a downstream effector of SOX9 (GSE69110) (*Ohba et al., 2015*). When SOX9 was overexpressed in fibroblasts, *GRASLND* expression was increased (~2 fold). We further confirmed this by utilizing SOX9 transgene overexpression in our MSCs culture (*Figure 3E*). Interestingly, although TGF-β3 has been demonstrated to act upstream of SOX9, exogenous addition of this growth factor alone did not result in enhanced *GRASLND* expression. It is notable that SOX9 levels in GFP controls were indistinguishable between TGF-β3 conditions at the time of investigation (1 week in monolayer culture), consistent with our previous finding that SOX9 was not upregulated until later timepoints in MSC chondrogenesis (*Huynh et al., 2018b*). Therefore, TGF-β3, despite being a potent growth factor, is not sufficient to elevate *GRASLND* expression. Instead, *GRASLND* appeared to be a downstream target of SOX9.

## Enhanced chondrogenesis for cartilage tissue engineering with *GRASLND*

As knockdown of *GRASLND* inhibited GAG and collagen deposition, we investigated whether overexpression of *GRASLND* would enhance chondrogenesis. We assessed this question by both transgene ectopic expression and by CRISPR-dCas9 (Clustered regularly interspaced short palindromic repeats – catalytically dead Cas9) mediated in-locus activation.

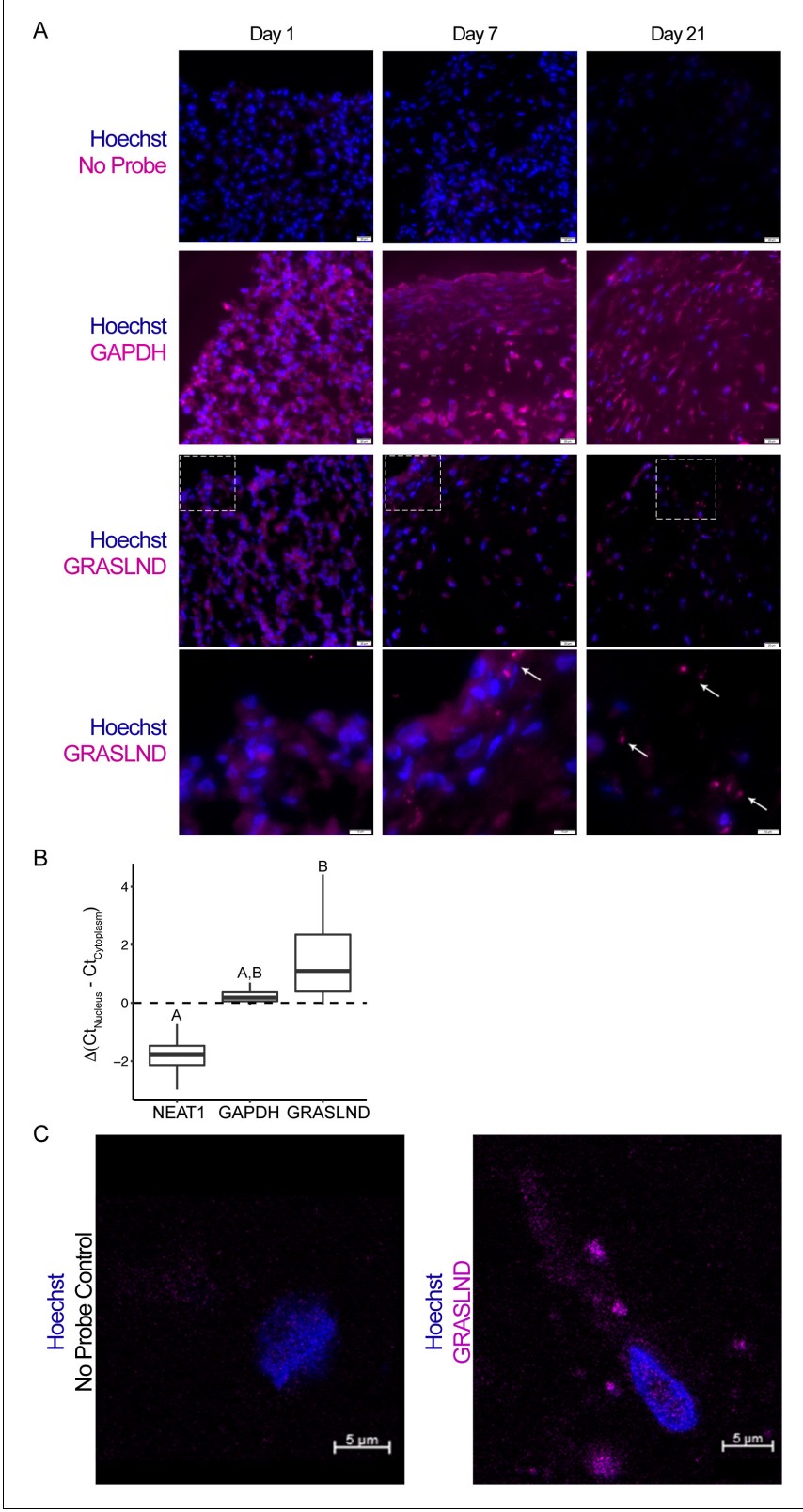

**Figure 2.** *GRASLND* is localized to the cytoplasm. (**A**) RNA in situ hybridization of MSC-derived pellets at different time points during chondrogenesis. *GAPDH* and *GRASLND* probes were hybridized on separate slides. Top three panels, scale bar = 20 μm; bottom panel, scale bar = 10 μm. (**B**) qRT-PCR of the nuclear versus cytoplasmic fraction of day 21 MSC pellets (n = 4). *NEAT1, Nuclear Paraspeckle Assembly Transcript 1.* One-way ANOVA

*Figure 2 continued on next page*

*Figure 2 continued*
followed by Tukey post-hoc test ($\alpha = 0.05$) was used. Groups of different letters are statistically different from one another. (C) Confocal microscopy on MSC-derived pellets. Scale bar = 5 µm.
The online version of this article includes the following source data for figure 2:

**Source data 1.** Cloning primer sequences (5′ – sequence – 3′).
**Source data 2.** *GRASLND* probe set sequences (5′ – sequence – 3′).

We designed our lentiviral transfer vector to carry a BGH-pA (Bovine Growth Hormone Polyadenylation) termination signal downstream of *GRASLND* to allow for its correct processing (*Figure 4—figure supplement 1A*). In addition, *GRASLND* was also driven under a doxycycline-inducible promoter, enabling the temporal control of its expression. We utilized this feature to induce *GRASLND* only during chondrogenic culture (*Figure 4A*). This experimental design focused solely on the role of

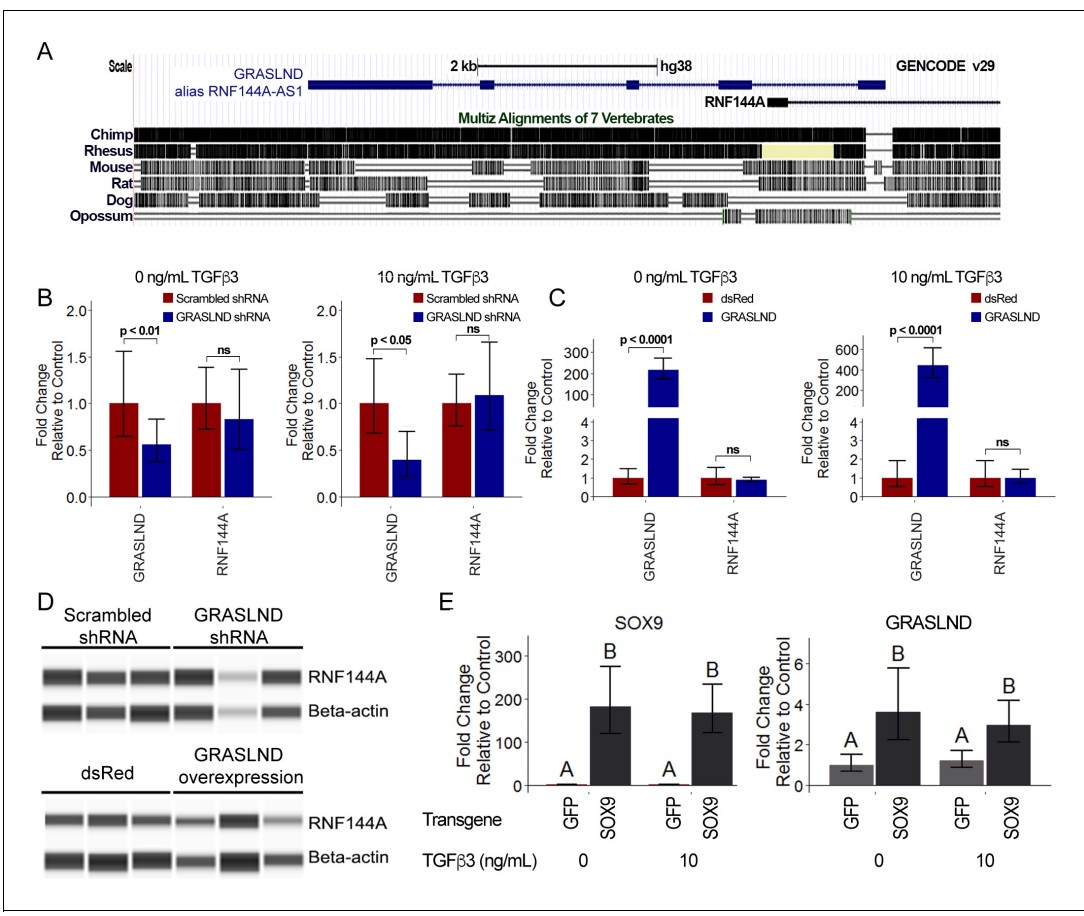

**Figure 3.** *GRASLND* relationship to RNF144A and SOX9. (A) *GRASLND* genomic location and conservation across different species. Data were retrieved from UCSC Genome Browser. (B) Knockdown of *GRASLND* and expression of *RNF144A* (n = 4). (C) Overexpression *GRASLND* and expression of *RNF144A* (n = 4). Welch's t-test. (D) Protein amount of RNF144A by western blot in variation of *GRASLND* levels. Lanes indicate biological replicates. Full bands are shown in *Figure 3—figure supplement 1*. (E) *GRASLND* level in GFP- or SOX9-transduced MSCs under different doses of TGF-β3 (n = 6). Two-way ANOVA followed by Tukey post-hoc test ($\alpha = 0.05$) was carried out on the effect of SOX9 overexpression (p<0.0001) and doses of TGF-β3 (p>0.05). The interaction between two tested factors (SOX9 overexpression and TGF-β3 doses) was not significant (p>0.05). Groups of different letters are statistically different. ns, not significant.
The online version of this article includes the following figure supplement(s) for figure 3:

**Figure supplement 1.** Full bands of western blot from *Figure 3D*.

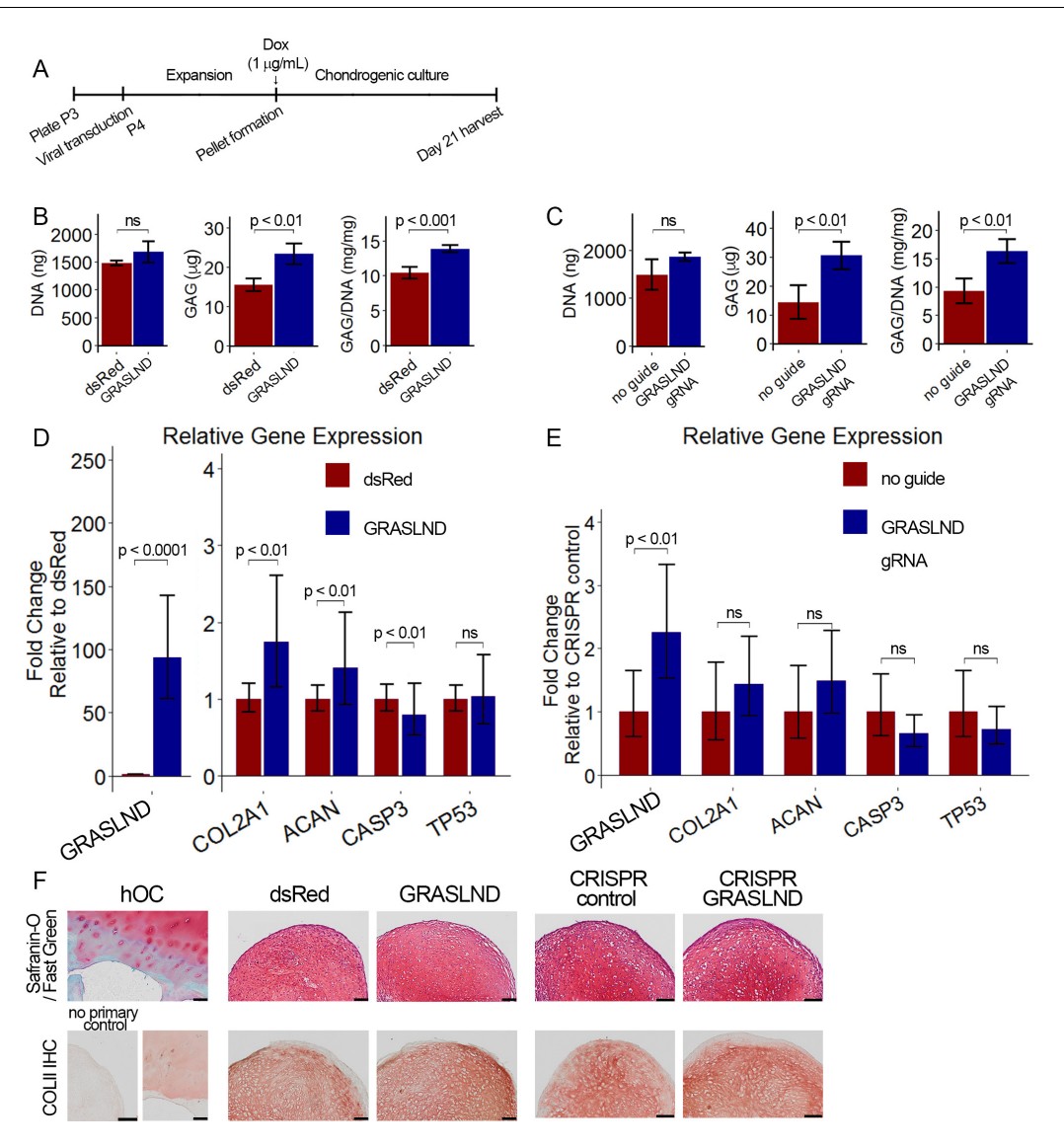

**Figure 4.** *GRASLND* enhances chondrogenesis. (A) Experimental timeline. (B, C) Biochemical analyses of day 21 MSC pellets (n = 4). Welch's t-test. (D, E) qRT-PCR analyses of day 21 MSC pellets (n = 5 in panel [D]; n = 6 in panel [E]). Welch's t-test. (F) Representative histological images of day 21 MSC pellets. COLII IHC, collagen type II immunohistochemistry; hOC, human osteochondral control. Scale bar = 100 μm. (B, D, F) Transgene ectopic expression of *GRASLND*. (C, E, F) CRISPR-dCas9-VP64-induced activation of *GRASLND*. ns, not significant (p>0.05).

The online version of this article includes the following figure supplement(s) for figure 4:

**Figure supplement 1.** Effect of GRASLND overexpression across time points and tested doses.
**Figure supplement 2.** Synthetic guide RNA screening for efficient activation of endogenous GRASLND.

---

*GRASLND* during chondrogenesis, while successfully eliminating its effect in MSC maintenance and expansion from our analysis. As control, a vector encoding the *Discosoma* sp. red fluorescent protein (dsRed) coding sequence in place of *GRASLND* was utilized. As doxycycline was most potent at 1 μg/mL (*Figure 4—figure supplement 1B,C*), this dose was used for all of the following experiments.

To determine whether *GRASLND* would improve chondrogenesis at lower doses of growth factor or at earlier time points, we compared DNA and GAG levels from pellets cultured under different TGF-β3 concentrations on day 7, day 14, and day 21 (*Figure 4—figure supplement 1D–F*). In agreement with our knockdown data, DNA content was unaffected. On the other hand, increases in GAG

were observed at higher doses and at later time points, especially at 10 ng/mL of TGF-β3. It appears that an elevated level of *GRASLND* alone was not sufficient to enhance GAG deposition when lower levels of TGF-β3 were used (0.1 ng/mL and 1 ng/mL) or at earlier time points (day 7 and day 14), and that *GRASLND* may act in concert with other downstream effectors, which were not present at lower doses of TGF-β3 or at earlier time points in the process.

When chondrogenesis was induced with 10 ng/mL of TGF-β3 and assessed at 21 days post induction, overexpression of *GRASLND* resulted in higher amounts of GAG deposition (p<0.001) (*Figure 4B*), consistent with our data on the gene expression level (*Figure 4D*). We observed a slight increase in chondrogenic markers (*COL2A1*, *ACAN*), and a slight decrease in the apoptotic marker *CASP3*, whereas cellular senescence was not different between the two groups (*TP53*) (*Figure 4D*). Histologically, pellets derived from dsRed-transduced MSCs exhibited normal GAG and collagen type II staining, indicating successful chondrogenesis. The control pellets were indistinguishable from those derived from *GRASLND*-transduced MSCs (*Figure 4F*), albeit macroscopically smaller at the time of harvest.

These findings were further confirmed using CRISPR-dCas9-VP64-mediated activation of endogenous *GRASLND*. This system had been previously utilized to upregulate various transcription factors that efficiently induce embryonic fibroblasts into neurons (*Black et al., 2016*; *Perez-Pinera et al., 2013*). After screening eleven synthetic gRNAs, we selected the one with the highest activation level (*Figure 4—figure supplement 2*). When *GRASLND* was transcriptionally activated with CRISPR-dCas9, chondrogenesis was enhanced as evidenced by an elevated amount of GAG deposition (p<0.01); DNA amount may also be slightly increased, albeit not to a statistically significant level (*Figure 4C*). Similar trends were detected by qRT-PCR (*Figure 4E*) and histology (*Figure 4F*). It is worth noting that CRISPR-dCas9-mediated activation only resulted in a moderate upregulation of *GRASLND* expression relative to transgene ectopic expression (2-fold vs 100-fold). However, the functional outcome was more pronounced with CRISPR-dCas9. We observed an approximately 50% increase in the level of GAG produced when normalized to DNA (9.4 ± 2.19 mg/mg vs 16.3 ± 2.08 mg/mg), compared to 30% detected with ectopic expression (10.5 ± 0.84 mg/mg vs 13.9 ± 0.52 mg/mg).

## *GRASLND* inhibits type II interferon signaling potentially by binding to EIF2AK2 and protects engineered cartilage from interferon

To decipher the potential signaling pathways involved, we chondrogenically induced MSCs in the presence or absence of *GRASLND*, and then utilized RNA-seq to compare the global transcriptomic changes between two conditions. As expected, *GRASLND* depletion resulted in impaired expression of chondrocyte-associated genes such as *TRPV4* and *COL9A2* (top 20 downregulated genes ranked by adjusted p-values) (*Figure 5A*). Skeletal system development and extracellular matrix organization were among the pathways most affected by the knockdown (*Figure 5B*). Surprisingly, pathways pertaining to interferon response were highly enriched in the upregulated gene list upon silencing of *GRASLND*. The top 20 upregulated genes involved many IFN downstream targets (*MX2*, *IFI44*, *IFI44L*, *IFITM1*, *IFI6*, *IFIT1*, *STAT1*, *MX1*, *IFIT3*, *OAS3*, *OAS2*), with both type I (IFN-α, IFN-β) and type II (IFN-γ) found to be enriched in our gene ontology analysis (*Figure 5B*). Furthermore, upregulated genes were also found to exhibit DNA-binding motifs for transcription factors of the IFN pathways: STAT1, STAT2, IRF1, and IRF2 (*Table 2*). A full list of differentially expressed genes is provided in *Supplementary file 2*. Further bioinformatic analyses created a network of potential transcription regulators as well as gene ontology terms for the upregulated gene cohort as a result of *GRASLND* silencing (*Figure 5C*). Taken together, *GRASLND* may act to suppress the activities of these transcription factors, and as a result could affect IFN signaling pathways during chondrogenesis.

To further confirm this relationship, we performed luciferase reporter assays for interferon signaling upon *GRASLND* knockdown. Utilizing specific reporter constructs, we were able to determine whether *GRASLND* acted on type I or type II IFN. Our results indicated that a decreased level of *GRASLND* led to a heightened type II (IFN-γ) (*Figure 5E*) response but not to a heightened type I (IFN-β) response (*Figure 5D*). Importantly, luminescence activities between scrambled control and *GRASLND* knockdown were indistinguishable from each other in basal, IFN-free conditions. This indicates that at the basal level, the two groups responded similarly to lentiviral transduction, and that the observed difference in IFN signal was a consequence of *GRASLND* downregulation.

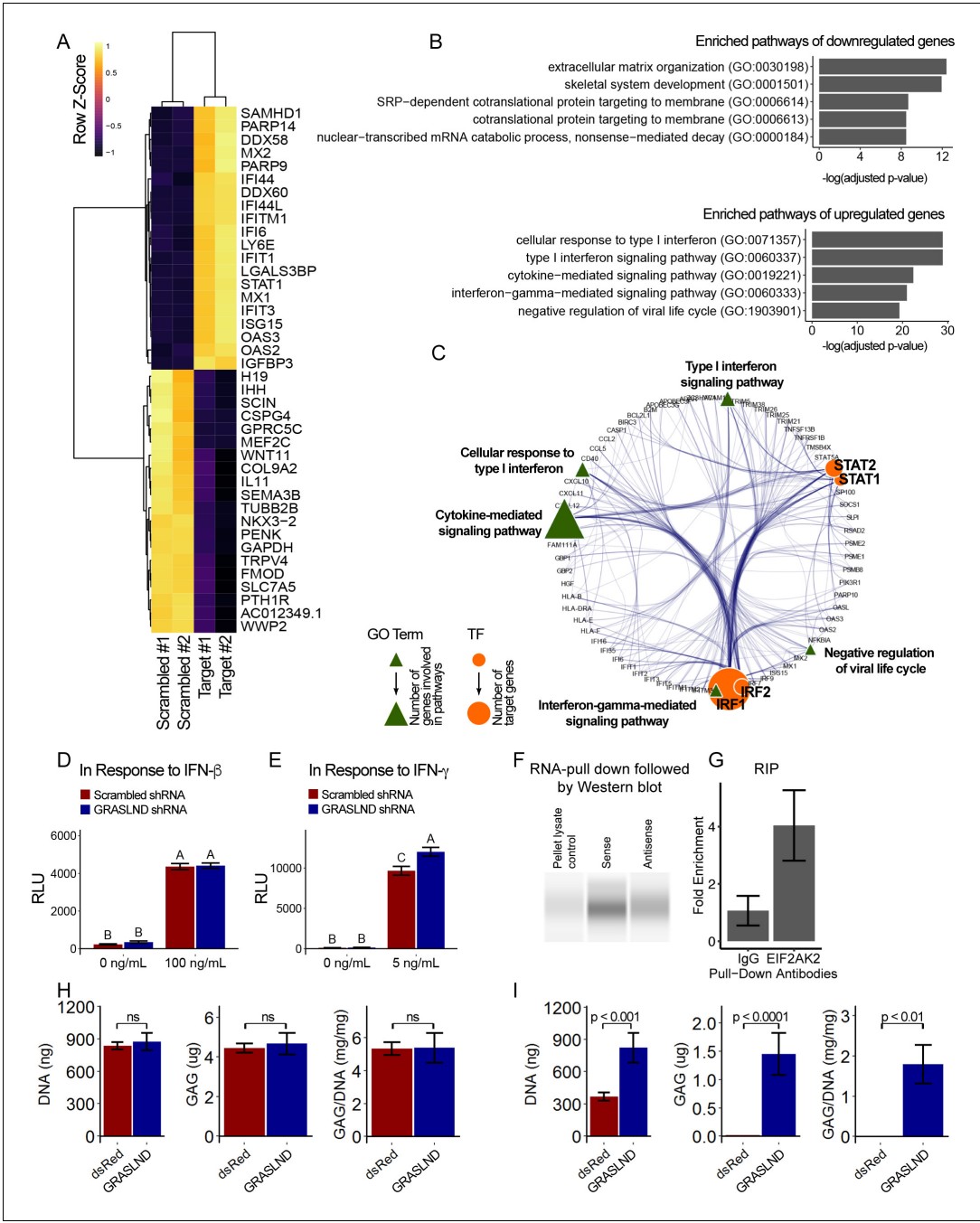

**Figure 5.** *GRASLND* suppresses interferon type II signaling. (**A**) Top 20 up- and down-regulated genes in *GRASLND* KD pellets compared to scrambled controls. (**B**) Gene ontology analysis of affected pathways. (**C**) Upregulated targets and related gene ontology terms and potential transcription factors. (**D,E**) Luciferase reporter assays on MSCs transduced with: (**D**) ISRE promoter element (n = 3), or (**E**) GAS promoter element (n = 3). Two-way ANOVA followed by Tukey post-hoc test (α = 0.05). Groups of different letters are statistically different. (**F**) RNA pull-down followed by western blot (full bands are shown in *Figure 5—figure supplement 2*). (**G**) RNA immunoprecipitation confirmed EIF2AK2 as the binding partner of GRASLND (n = 2). (**H**) Biochemical assays on MSC-derived pellets cultured under 100 ng/mL of IFN-β (n = 4). (**I**) Biochemical assays on MSC-derived pellets cultured under 5 ng/mL of IFN-γ (n = 6). Welch's t-test. ns, not significant.

The online version of this article includes the following figure supplement(s) for figure 5:

**Figure supplement 1.** Relationship between GRASLND and IFN.

**Figure supplement 2.** Full bands of RNA pull-down followed by western blot.

**Table 2.** Top 5 enriched Cis-BP motifs and associated transcription factors for upregulated genes upon GRASLND knockdown.

| Transcription factor | Cis-BP motif* | Number of genes with enriched motifs/number of upregulated genes |
|---|---|---|
| STAT2 | cisbp__M4635 | 212/817 |
| IRF2 | cisbp__M6308 | 189/817 |
| IRF1 | cisbp__M6307 | 220/817 |
| IRF1 | cisbp__M1882 | 153/817 |
| STAT1 | cisbp__M4707 | 262/817 |

*Cis-BP: Catalogue of Inferred Sequence Preferences of DNA-Binding Proteins (**Weirauch et al., 2014**). Curated position weight matrices were retrieved from http://motifcollections.aertslab.org.

Since *GRASLND* was expressed in the cytoplasm (*Figure 2C*), we hypothesized that it is part of an RNA–protein complex. To test this, we performed an RNA pull-down assay, followed by mass spectrometry. Here, streptavidin beads were used as control, or conjugated to sense or antisense strands of *GRASLND*. Naked or conjugated beads were then incubated with lysates from day 21 pellets, from which bound proteins were eluted for further analyses. We found that Interferon-Induced Double-Stranded RNA-Activated Protein Kinase (EIF2AK2) peptides were detected at elevated levels in sense samples as compared to antisense controls ($p < 0.05$); peptides were undetected in naked bead controls. Subsequent RNA pull-down followed by western blot confirmed EIF2AK2 as a binding partner of *GRASLND* (*Figure 5F*). We detected an increased level of EIF2AK2 bound to the sense strand of *GRASLND* relative to the antisense strand or the pellet lysate control. Similarly, *GRASLND* was found to be associated with endogenous EIF2AK2 by RNA immunoprecipitation (RIP) (*Figure 5G*). On the basis of these findings, we speculate that this association of *GRASLND* RNA to EIF2AK2 could potentially result in downregulation of IFN-γ signaling.

Interestingly, by mining a published microarray database (GSE57218) (*Ramos et al., 2014*), we found that IFN-related genes (*STAT1*, *IFNGR2*, *NCAM1*, *MID1*) were highly elevated in the cartilage tissues of osteoarthritis patients (*Figure 5—figure supplement 1A*). As the microarray did not contain probes for *GRASLND*, no information on its expression could be extracted. In addition, we identified another independent study that reported changes in the transcriptomes of intact and damaged cartilage tissues (E-MTAB-4304) (*Dunn et al., 2016*). Similarly, a cohort of IFN-related genes was also upregulated in damaged cartilage, especially *STAT1* and *IFNGR1* (*Figure 5—figure supplement 1B*). Interestingly, we identified a negative correlation between *GRASLND* and a few IFN related genes (*IFNGR1*, *ICAM1*) in damaged cartilage (*Figure 5—figure supplement 1C*). Therefore, we proposed that *GRASLND* may possess some therapeutic potential through suppression of IFN signaling in osteoarthritis. To evaluate this possibility, we implemented the use of the *GRASLND* transgene in engineered cartilage cultured under IFN addition (100 ng/mL of IFN-β or 5 ng/mL of IFN-γ). We determined doses of IFN-β and IFN-γ by selecting the lowest concentration at which day 21 pellets exhibited GAG loss when compared to no IFN control. Consistent with luciferase reporter assays, the protective effect of *GRASLND* was observed upon IFN-γ challenge but not upon IFN-β challenge (*Figure 5H,I*). However, we observed a reduced level of GAG production compared to normal conditions, suggesting that *GRASLND* can protect the ECM from degradation, but not completely to control levels.

### *GRASLND* enhanced the chondrogenesis of adipose-derived stem cells

To determine whether the function of *GRASLND* is unique to MSCs or present in other adult stem cells, we addressed whether modulating *GRASLND* expression could also improve chondrogenesis of adipose stem cells (ASCs). We observed an increase in GAG production when *GRASLND* was overexpressed in ASCs compared to control ($p<0.0001$) (*Figure 6A*), although *ACAN* levels were not significantly increased. Importantly, *COL2A1* expression was significantly elevated (~5 fold) with overexpression of *GRASLND* (*Figure 6B*). On the basis of these data, it appears that *GRASLND* uses the same mechanism across these two cell types, asserting a pan effect on potentiating their chondrogenic capabilities. It is worth noting that histologic examination of the engineered cartilage

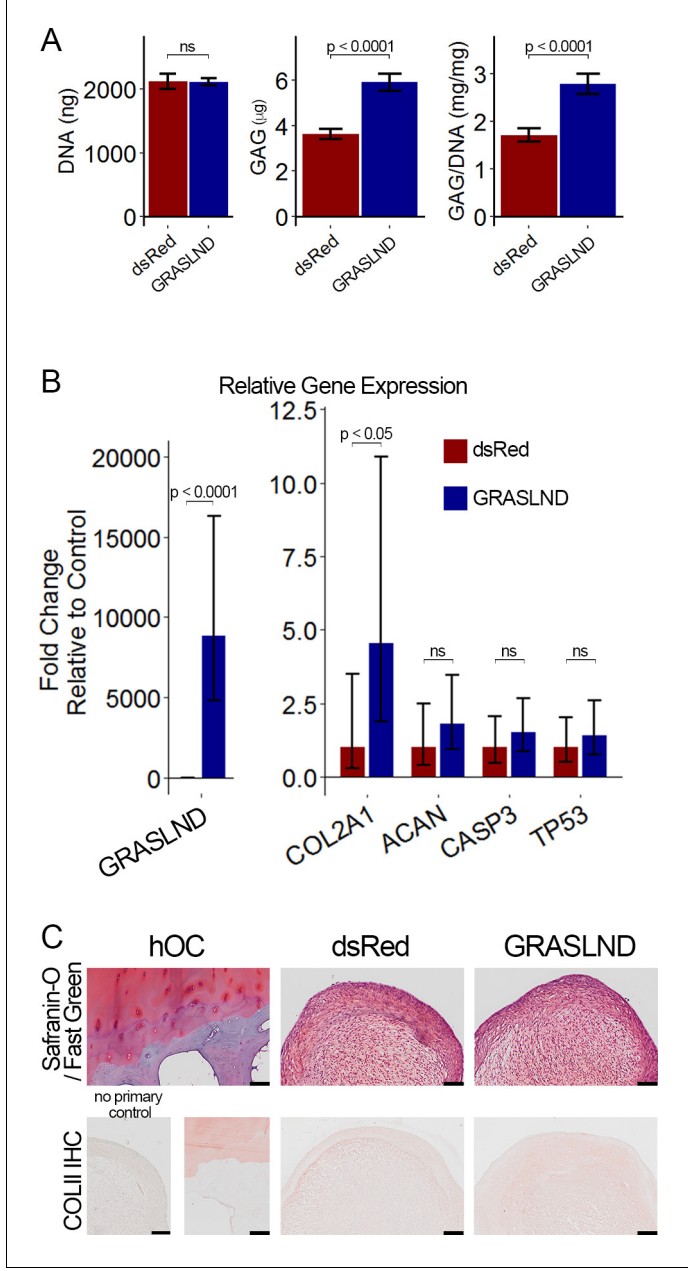

**Figure 6.** *GRASLND* enhances chondrogenesis in adipose-derived stem cells. (**A**) Biochemical analyses (n = 5). (**B**) qRT-PCR analyses (n = 6). (**C**) Representative histological images of day 21 ASC pellets. COLII IHC, Collagen type II immunohistochemistry; hOC, Human osteochondral control. Scale bar = 100 μm. Welch's t-test. ns, not significant.

showed a similar level of collagen type II in pellets with *GRASLND* overexpression compared to the dsRed control (*Figure 6C*), suggesting that the influence of *GRASLND* could be variable in different target cells.

## Discussion

Here, we identified and demonstrated the first functional study of lncRNA *GRASLND,* which acts to enhance stem cell chondrogenesis. Knockdown of *GRASLND* via shRNA inhibited chondrogenesis, whereas ectopic transgene or CRISPR-based overexpression of *GRASLND* enhanced chondrogenesis of MSCs and ASCs. Pathway analysis revealed a link between *GRASLND* and the IFN-γ signaling pathway in this process, which was confirmed by the identification of EIF2AK2 as a *GRASLND* binding partner. Unfortunately, lack of a known murine homolog makes it difficult to study *GRASLND* in vivo, and thus future studies may require *GRASLND* transgenic models in primate species.

In the context of the musculoskeletal system, IFN is mostly recognized for its role in bone development and homeostasis (*Dieudonne et al., 2013*; *Rostovskaya et al., 2018*; *Takayanagi et al., 2002a*; *Takayanagi et al., 2002b*; *Li, 2013*; *Xiao et al., 2004*) and myogenesis (*Jang and Baik, 2013*; *Cheng et al., 2008*; *Londhe and Davie, 2011*), as well as for its crosstalk with TGF-β in wound healing (*Ishida et al., 2004*). Notably, IFN-γ has been suggested to inhibit collagen synthesis in dermal fibroblasts, myofibroblasts, and articular chondrocytes (*Ishida et al., 2004*; *Yufit et al., 1995*; *Harrop et al., 1995*; *Granstein et al., 1990*; *Amento et al., 1985*). Furthermore, the JAK/STAT pathway, which involves IFN downstream effectors, has also been shown to inhibit chondrocyte proliferation and differentiation (*Sahni et al., 1999*; *Sahni et al., 2001*). Here, we found that *GRASLND* acts to suppress the IFN mechanism. In addition, we also present evidence that indicates an interaction between *GRASLND* and EIF2AK2 (also referred to as PKR). Canonically, a crucial player in protein synthesis, EIF2AK2, has also been reported to control STAT signaling by directly binding to and preventing its association with DNA for gene activation (*Wang et al., 2006*; *Wong et al., 1997*). In addition, several studies have suggested that highly structured, single-stranded RNA can also activate PKR EIF2AK2 via its double-stranded RNA-binding domains (dsDRBs) (*Osman et al., 1999*; *Ben-Asouli et al., 2002*; *Cohen-Chalamish et al., 2009*; *Nallagatla et al., 2007*; *Mayo and Cole, 2017*). Our RNA-seq data suggested that upon *GRASLND* knockdown, a cohort of downstream targets of STATs were upregulated. On the basis of the presence of DNA-binding motifs in the investigated targets, we identified both STAT1 and STAT2 as potential regulators of genes that are disrupted by *GRASLND* knockdown. However, our luciferase reporter assays pointed towards a mechanism in IFN type II (gene activation by STAT1 homodimer) pathways rather than type I (gene activation by STAT1/STAT2 heterodimer) pathways. Thus, we hypothesized that *GRASLND* could form a secondary structure to bind and activate EIF2AK2, which in turn inhibits STAT1-related transcriptional function. This mechanism supports the hypothesis that modulation of IFN-γ via the JAK/STAT pathway, achieved by the *GRASLND*–EIF2AK2 RNA–protein complex, is important for cellular proliferation and differentiation during chondrogenesis.

Upregulation of IFN has also been implicated in arthritis by several studies (*Boissier et al., 1995*; *Cooper et al., 1988*; *Westacott et al., 1990*; *Kahle et al., 1992*). Publicly available databases provide evidence corroborating similar patterns of IFN in degenerated cartilage (*Ramos et al., 2014*). As *GRASLND* inhibits IFN, utilization of this lncRNA offers potential in both MSC cartilage tissue engineering and OA treatment. As a proof of concept, we showed that *GRASLND* could enhance matrix deposition across cell types of origin, with and without interferon challenge in vitro. Future studies may wish to investigate whether *GRASLND* can protect cartilage from degradation in a milieu of pro-inflammatory cytokines in vivo.

Since lentivirus was used to manipulate the expression of *GRASLND*, it is possible that our observations were confounded by the cellular response to viral infection. However, our luciferase reporter assays demonstrated that basal luminescence levels (with no interferon supplementation) in the scrambled controls and the shRNA treatments were indistinguishable. This finding suggests that altered levels of interferon signaling can be attributed to experimentally varied levels of *GRASLND* and not to the presence of lentivirus. Our data indicate that *GRASLND* acts through type II rather than type I IFN. We found that 5 ng/mL of IFN-γ was still more detrimental to chondrogenic constructs than 100 ng/mL of IFN-β. One potential explanation for this phenomenon may be the skewed

distribution of available surface receptors between type I and type II IFN (IFNAR vs IFNGR). Indeed, MSCs express a much lower level of *IFNAR2* than of *IFNAR1*, *IFNGR1*, or *IFNGR2* (both in GSE109503 [*Huynh et al., 2018b*] and in GSE129985 [this manuscript]). As these receptors function as heterodimers (*Brierley and Fish, 2002*; *Hu and Ivashkiv, 2009*), response to type I may be stunted due to IFNAR2 deficiency.

Furthermore, we showed that a modified CRISPR-dCas9 system could be used successfully for endogenous transcriptional activation of lncRNA. This system had been previously used in other cell types to regulate the expression of both protein-coding and non-coding genes (*Black et al., 2016*; *Perez-Pinera et al., 2013*; *Bester et al., 2018*; *Liu et al., 2017*). We showed that CRISPR may be more effective than transgene expression, as indicated by a larger increase in GAG production, despite lower levels of overall gene activation. As *GRASLND* does not regulate RNF144A, it is evident that *GRASLND* acts in *trans*. However, we speculate that the CRISPR-dCas9 system could also be useful for gain-of-function studies to investigate lncRNAs acting in *cis*, as well as for studies of lncRNAs that are difficult to obtain via molecular cloning because of their secondary structures, highly repeated sequence or GC-rich content.

In conclusion, we have identified *GRASLND* as an important regulator of MSC chondrogenesis. *GRASLND* acts downstream of SOX9 and enhances cartilage-like matrix deposition in stem cell-derived constructs. Moreover, *GRASLND* functions to suppress IFN via EIF2AK2, and as a result induces adult stem cells towards a more chondrocyte-like lineage. It is likely that the *GRASLND*–EIF2AK2 RNA–protein complex inhibits STAT1 transcriptional activity. These findings suggest that *GRASLND* has potential utility in enhancing stem cell chondrogenesis for therapeutic applications such as cartilage tissue engineering or for the treatment of OA.

## Materials and methods

### Cell culture

Bone marrow was obtained from discarded and de-identified waste tissue from adult bone marrow transplant donors in accordance with the Institutional Review Board of Duke University Medical Center. Adherent cells were expanded and maintained in expansion medium: DMEM-low glucose (Gibco), 1% penicillin/streptomycin (Gibco), 10% fetal bovine serum (FBS) (ThermoFisher), and 1 ng/mL basic fibroblast growth factor (Roche) (*Hagmann et al., 2013*).

Adipose-derived stem cells (ASCs) were purchased from ATCC (SCRC-4000) and cultured in complete growth medium: mesenchymal stem cells basal medium (ATCC PCS-500–030), mesenchymal stem cell growth kit (ATCC PCS-500–040) (2% FBS, 5 ng/mL basic recombinant human FGF, 5 ng/mL acidic recombinant human FGF, 5 ng/mL recombinant human EGF, 2.4 nM L-alanyl-L-glutamine), and 0.2 mg/mL G418.

### Plasmid construction
#### shRNA
Short hairpin RNA (shRNA) sequences for specific genes of interest were designed with the Broad Institute GPP Web Portal (*Moffat et al., 2006*). For each gene, six different sequences were selected for screening, after which the two most effective were chosen for downstream experiments in chondrogenic assays. Selected shRNAs were cloned into a modified lentiviral vector (Addgene #12247) using MluI and ClaI restriction sites, as described previously (*Diekman et al., 2015*). A complete list of effective shRNA sequences is presented in *Figure 1—source data 1*.

### Transgene overexpression of *GRASLND*
A derivative vector from modified TMPrtTA (*Glass et al., 2014*; *Barde et al., 2006*) was created with NEBuilder HiFi DNA Assembly Master Mix (New England Biolabs). Backbone was digested with EcoRV-HF (New England Biolabs) and PspXI (New England Biolabs). The following resultant fragments were amplified by polymerase chain reaction and assembled into the digested plasmid: Tetracycline-responsive element and minimal CMV promoter (TRE/CMV), firefly luciferase, bGH poly(A) termination signal (BGHpA). Primers and plasmids for cloning are provided in *Figure 2—source data 1*.

The full sequence of *GRASLND* transcript variant 1 (RefSeq NR_033997.1) was synthesized by Integrated DNA Technologies, Inc. *GRASLND* or the *Discosoma* sp. red fluorescent protein coding sequence (dsRed) were cloned into the above derivative tetracycline-inducible plasmid with NEBuilder HiFi DNA Assembly Master Mix (New England Biolabs) at NheI and MluI restriction sites (pLVD-GRASLND and pLVD-dsRed). Amplifying primers are listed in *Figure 2—source data 1*.

## CRISPR-dCas9 activation of *GRASLND*

Guide RNA sequences were designed using the UCSC genome browser (http://genome.ucsc.edu/) (*Kent et al., 2002*), integrated with the MIT specificity score calculated by CRISPOR and the Doench efficiency score (*Doench et al., 2016*; *Haeussler et al., 2016*). Oligonucleotides (Integrated DNA Technologies, Inc) were phosphorylated, annealed, and ligated into the pLV-hUbC-dCas9-VP64 lentiviral transfer vector (Addgene #53192) previously digested at BsmBI restriction sites (*Kabadi et al., 2014*). Eleven potential guide RNA sequences were selected and screened for their efficacy, and the gRNA with the highest activation potential was chosen for further experiments (*Figure 4—figure supplement 2*). The synthetic gRNA used in all CRISPR-dCas9 activation experiments has the following sequence: 5′-CCACTGGGGATAGTTCCCTG-3′.

## Lentivirus production

HEK 293T producer cells were maintained in 293T medium: DMEM-high glucose (Gibco), 10% heat inactivated FBS (Atlas), and 1% penicillin/streptomycin (Gibco). To produce lentivirus for pellet studies, HEK 293T cells were plated at $3.8 \times 10^6$ cells per 10 cm dish (Corning) or at $8.3 \times 10^6$ cells per 15 cm dish (Falcon) in 293T medium. The following day, cells were co-transfected by calcium phosphate precipitation with the appropriate transfer vector (20 μg for 10 cm dish; 60 μg for 15 cm dish), the second-generation packaging plasmid psPAX2 (Addgene #12260) (15 μg for 10 cm dish; 45 μg for 15 cm dish), and the envelope plasmid pMD2.g (Addgene #12259) (6 μg for 10 cm dish; 18 μg for 15 cm dish). Cells were incubated at 37°C overnight. The following day, fresh medium consisting of DMEM-high glucose (Gibco), 10% heat-inactivated FBS (Atlas), 1% penicillin/streptomycin (Gibco), and 4 mM caffeine (Sigma-Aldrich) was exchanged (12 mL for 10 cm dish; 36 mL for 15 cm dish). Lentivirus was harvested 24 hr post medium change (harvest 1), when fresh medium was exchanged again. 48 hr post medium change, harvest two was collected. Harvest one and harvest two supernatants were pooled, filtered through 0.45 μm cellulose acetate filters (Corning), concentrated, aliquoted, and stored at −80°C for future use.

To produce lentivirus for shRNA and gRNA screening, HEK 293T cells were plated at $1.5–2 \times 10^6$ cells per well in a 6-well plate in DMEM-high glucose (Gibco), and 10% heat inactivated FBS (Atlas). The following day, cells were co-transfected with 2 μg of the appropriate transfer vector, 1.5 μg of the packaging plasmid psPAX2 (Addgene #12260), and 0.6 μg of the envelope plasmid (Addgene #12259) with Lipofectamine 2000 (ThermoFisher) following manufacturer's protocol. Harvest and storage were performed as described above.

For knockdown experiments, lentivirus was titered by determining the number of antibiotic-resistant colonies after puromycin treatment. For overexpression experiments, lentivirus was titered by measuring integrated lentiviral copy number in host DNA with qRT-PCR as previously described (*Sastry et al., 2002*). Control and tested groups were targeted at similar MOIs.

## Lentivirus transduction

Cells were plated at 4500 cells/ cm$^2$ for one day and then transduced with appropriate lentivirus in expansion medium supplemented with 4 μg/mL polybrene (Sigma-Aldrich). Twenty-four hours post transduction, cells were rinsed once in phosphate buffered saline (PBS). Cells were cultured with fresh medium exchange every three days.

## Cytotoxicity assay

Seven days post viral transduction, medium was collected and the amount of lactose dehydrogenase (LDH) was measured as indirect output for cellular toxicity. Assays were performed following manufacturer's protocol (Promega). Absorbance signal was recorded at 490 nm with the Cytation 5 instrument (BioTek).

## RNA-seq library preparation

Isolated RNAs were stored at −80°C and submitted to the Genome Technology Access Center at Washington University in St Louis for library preparation and sequencing on a HiSeq 2500 (2 × 101 bp). Libraries were prepared using TruSeq Stranded Total RNA with Ribo-Zero Gold kit (Illumina).

## RNA pull-down and mass spectrometry

The full sequence of GRASLND transcript variant 1 (RefSeq NR_033997.1) was synthesized by Integrated DNA Technologies, Inc, and cloned into the pGEM-T Easy Vector System (Promega) using the EcoRV site. This served as a template for subsequent in vitro transcription using the Riboprobe Combination Systems Kit (Promega), with spiked-in biotin RNA labeling mix (Roche). Resulted biotinylated sense and control antisense transcripts of *GRASLND* were stored at −80°C until further processing. Cell lysates from day 21 pellets were homogenized in mRIPA buffer (Cell Signaling) and centrifuged at 14,000 rpm for 15 min. The protein concentration of cell lysates was measured and adjusted to 2 mg/mL. 500 µL of total protein (1 mg) were incubated with either 1.5 µg of *GRASLND*-sense or -antisense RNA transcripts tagged with biotin-16-UTP overnight (12 hr). Following incubation, the RNA–protein mixtures and cell lysates (control) were incubated with 100 µL of prewashed streptavidin beads for 3 hr at 4°C (Pierce MS-Compatible Magnetic IP Kit, Streptavidin). The streptavidin beads were then washed five times in 800 µl of ice cold PBS. Beads were eluted twice, each with 30 µL of SDS elution buffer containing 100 mM Tris/HCl (pH 8), 4% SDS, and 50 mM DTT. The elution was used either for mass spectroscopy (Proteomics Core Facility, Washington University School of Medicine) or for Western blot (RayBiotech).

## RNA immunoprecipitation (RIP)

Day 21 pellets were harvested and stored at −80°C until further processing. Lysate was obtained by homogenizing day 21 pellets in 1.1 mL of complete RIP lysis buffer (1X RIP lysis buffer, 200X protease inhibitor cocktail, 400X RNase inhibitor) (Millipore) with a bead beater (BioSpec Products) at 2500 oscillations per minute for 3 min for a total of five times. The lysate was subsequently transferred to a new microcentrifuge tube, incubated on ice for 5 min to allow for cell swelling by the hypotonic RIP buffer, then stored at −80°C overnight.

RIP assay was performed using the EZ-Magna RIP Kit (Millipore) with the rabbit anti-PKR (alias for EIF2AK2) antibodies (Abcam) following manufacturer's protocol. Separation of beads during the procedure was carried out using the MiniMACS separator (Miltenyi Biotec). Briefly, magnetic beads were washed and prepared by incubating with 5 µg of rabbit anti-PKR antibodies or 5 µg of normal rabbit IgG (negative control) per RIP reaction with rotation. Once the beads were ready, 900 µL of fresh complete RIP immunoprecipitation buffer (1X RIP wash buffer, 0.5 M EDTA, 200X RNase inhibitor) was added to the magnetic beads, followed by 100 µL of pellet lysate per reaction. Tubes were incubated with rotation at 200 rpm overnight at 4°C (Benchmark Orbi-Sharker Jr). The next day, tubes were centrifuged briefly, and magnetic beads were washed with cold RIP wash buffer for a total of six times. Proteins were subsequently degraded from resulting pull-down with proteinase K (1X RIP wash buffer, 10% SDS, 8.3X proteinase K) at 55°C for 30 min. After the incubation, tubes were centrifuged briefly and beads were separated. Supernatant was transferred into a new microcentrifuge tube, to which 250 µL of buffer RL was added (Norgen Biotek). RNA was isolated as described below using the Norgen Total RNA Isolation Plus Micro Kit (Norgen Biotek) following the manufacturer's protocol. An equal amount of eluted RNA was subsequently used for reverse transcription, followed by qRT-PCR as described below.

## Bulk RNA-seq analysis
### Alignment and read assignment

Demultiplexed raw sequencing files were generated by the Genome Technology Access Center at Washington University in St Louis. Reads were processed with trimmomatic-0.36 (*Bolger et al., 2014*), aligned with STAR-2.6.0 (*Dobin et al., 2013*) and counted with featureCounts/Subread-1.6.1 (*Liao et al., 2014*).

### Differential expression analysis

Downstream differential expression analysis was performed using DESeq2-1.16.1 (*Love et al., 2014*) (abs[log2 fold change]>1 and adjusted p-values<0.05).

### Gene ontology analysis

Gene ontology analysis of dysregulated genes was performed with enrichR-1.0 (*Chen et al., 2013*; *Kuleshov et al., 2016*).

### Transcription factor identification

Potential transcription factors were identified on the basis of the presence of annotated DNA-binding motifs with RcisTarget-1.0.2 (*Aibar et al., 2017*). Annotation databases for the motifs in human transcription factors were previously compiled and can be downloaded at https://resources.aertslab.org/cistarget/. Cis-BP motifs were ranked by normalized enrichment score (NES), and the top five were reported in this paper.

### Identification of lncRNA candidates

GSE109503 is the dataset that profiles transcriptomic changes of MSC chondrogenesis, composed of six time points (day 0, day 1, day 3, day 7, day 14, and day 21) and three biological replicates. Raw sequencing files were downloaded from the GEO Omnibus, and processed as described above. Candidates were first restricted to those differentially expressed per day pair-wise (abs[log2 fold change]>1 and adjusted p-values<0.05) and of detectable abundance (TPM >1 in more than six samples across the dataset). lncRNAs whose transcripts were not analyzed for transcript support level (ENSEMBL TSL) were also excluded. For the surviving genes, Pearson correlation analysis was then performed on mean expression per day. Candidates were identified as those with Pearson correlation values >0.9 to all three investigated markers (ALCAM, VCAM1, ENG for MSC markers; COL2A1, ACAN, COMP for chondrogenic markers; SOX5, SOX6, SOX9 for SOX transcription factors). GSE69110 depicts the transcriptomic changes of fibroblasts in response to SOX9 expression levels (*Supplementary file 1*). Raw sequencing files were downloaded from the GEO Omnibus, and processed similarly. Genes that were expressed differentially between two conditions (SOX9 overexpression versus GFP control) were then identified (abs[log2 fold change]>1 and adjusted p-values<0.1). The shortlist of lncRNAs are the intersecting candidates between genes emerging from the above Pearson correlation analysis and dysregulated genes from this dataset.

## Microarray analysis

Microarray processed data was downloaded from the GEO Omnibus and differential expression analysis was performed with limma-3.34.6 (*Ritchie et al., 2015*).

## Mass spectrometry analysis

Scaffold-4.8.4 (Proteome Software Inc) was used to validate MS/MS-based peptide and protein identifications. Peptide identifications were accepted if they could be established as having a greater than 66.0% probability of achieving an FDR less than 1.0% by the Scaffold Local FDR algorithm. Protein identification was accepted if they could be established at a greater than 95.0% probability and contained at least one identified peptide. Protein probabilities were assigned by the Protein Prophet algorithm (*Nesvizhskii et al., 2003*). Proteins that contain similar peptides and could not be differentiated on the basis of MS/MS analysis alone were grouped to satisfy the principles of parsimony. To identify differentially bound proteins, one-tailed t-test was performed on sense samples compared to naked beads, and sense samples were compared to antisense samples.

## Chondrogenesis assay

MSCs or ASCs were digested in 0.05% trypsin-EDTA (Gibco), and trypsin was inactivated with 1.5X volume of expansion medium. Dissociated cells were centrifuged at 200 x g for 5 min, and supernatant was aspirated. Subsequently, cells were washed in pre-warmed DMEM-high glucose (Gibco) three times, and resuspended at $5 \times 10^5$ cells/mL in complete chondrogenic medium: DMEM-high glucose (Gibco), 1% penicillin/streptomycin (Gibco), 1% ITS+ (Corning), 100 nM dexamethasone (Sigma-Aldrich), 50 µg/mL ascorbic acid (Sigma-Aldrich), 40 µg/mL L-proline (Sigma-Aldrich), and 10

ng/mL rhTGF-β3 (R and D Systems). 500 µL of the above cell mixture was dispensed into 15 mL conical tubes and centrifuged at 200 x g for 5 min. Pellets were cultured at 37°C in 5% $CO_2$ for 21 days with medium exchange every three days.

## Osteogenesis and adipogenesis assays

MSCs were plated at $2 \times 10^4$ cells/well in 6-well plates (Corning) and cultured for 4 days in MSC expansion medium, followed by induction medium for 7 days. Osteogenic induction medium includes: DMEM-high glucose (Gibco), 10% FBS, 1% penicillin/streptomycin (Gibco), 10 nM dexamethasone (Sigma-Aldrich), 50 µg/mL ascorbic acid (Sigma-Aldrich), 40 µg/mL L-proline (Sigma-Aldrich), 10 mM β-glycerol phosphate (Chem-Impex International), and 100 ng/mL rh-BMP2 (ThermoFisher). Adipogenic induction medium includes: DMEM-high glucose (Gibco), 10% FBS (ThermoFisher), 1% penicillin/streptomycin (Gibco), 1% ITS+ (Corning), 100 nM dexamethasone (Sigma-Aldrich), 450 µM 3-isobutyl-1-methylxanthine (Sigma-Aldrich), and 200 µM indomethacin (Sigma-Aldrich).

## Biochemical assays

Harvested pellets were stored at −20°C until further processing. Collected samples were digested in 125 µg/mL papain at 60°C overnight. A DMMB assay was performed as previously described to measure GAG production (Farndale et al., 1986). PicoGreen assay (ThermoFisher) was performed to measure DNA content following manufacture's protocol.

## Immunohistochemistry and histology

Harvested pellets were fixed in 4% paraformaldehyde for 48 hr, and processed for paraffin embedding. Samples were sectioned at 10 µm thickness, and subjected to either Safranin O – Fast Green standard staining (Estes et al., 2010) or to immunohistochemistry of collagen type II (Developmental Studies Hybridoma Bank, University of Iowa). Human osteochondral sections were stained simultaneously to serve as a positive control. Sections with no primary antibodies were used as negative control for immunohistochemistry.

## Single-molecule RNA fluorescence in situ hybridization (RNA FISH)

Harvested pellets were snap frozen in Tissue-Plus O.C.T. Compound (Fisher HealthCare) and stored at −80°C until further processing. Samples were sectioned at 5 µm thickness and slides were stored at −80°C until staining. Probe sets for RNA FISH were conjugated with Quasar 670 dye and were synthesized by LGC Biosearch Technologies to detect signal from a congregation of multiple probes binding to target DNA. GAPDH probe set was pre-designed by the manufacturer. Probe sets are listed in Figure 2—source data 2. Staining was carried out according to the manufacturer's protocol for frozen tissues. Slides were mounted with Prolong Gold anti-fade mountant with DAPI (ThermoFisher) and imaged with the Virtual Slide Microscope VS120 (Olympus) at lower magnification. Confocal microscopy (Zeiss LSM 880) was used to capture images at higher magnification with the Plan-Apochromat 63x/1.40 Oil DIC M27 objective. Fluorescence signal from target RNA FISH probes was captured using a 633 nm excitation wavelength coupled with the Airyscan detector (Zeiss) to achieve the best resolution with improved signal-to-noise ratio (Weisshart, 2014). Hoechst signal was captured on the PMT detector utilizing a 405 nm excitation wavelength.

## RNA isolation and quantitative RT-PCR

Norgen Total RNA Isolation Plus Micro Kits (Norgen Biotek) were used to extract RNA from pellet samples, and Norgen Total RNA Isolation Plus Kits (Norgen Biotek) were used for all other RNA isolation. For monolayers, cells were lysed in buffer RL and stored at −20°C until further processing. For pellets, harvested samples were snap frozen in liquid nitrogen and stored at −80°C until further processing. On the day of RNA isolation, pellets were homogenized in buffer RL using a bead beater (BioSpec Products) at 2500 oscillations per minute for 20 s for a total of three times. Subsequent steps were performed following the manufacturer's protocol.

Nuclear and cytoplasmic fractions from day 21 MSC pellets were separated with the NE-PER Nuclear and Cytoplasmic Extraction Reagents (ThermoFisher) following the manufacturer's protocol. The resulting extracts were immediately subjected to RNA isolation using Norgen Total RNA

Isolation Plus Micro Kits (Norgen Biotek) by adding 2.5 parts of buffer RL to 1 part of extract. Subsequent steps were carried out following the manufacturer's protocol.

Reverse transcription by Superscript VILO cDNA master mix (Invitrogen) was performed immediately following RNA isolation. cDNA was stored at −20°C until further processing. qRT-PCR was carried out using Fast SyBR Green master mix (Applied Biosystems) following the manufacturer's protocol. A complete list of primer pairs (synthesized by Integrated DNA Technologies, Inc) is reported in *Figure 1—source data 2*.

### Luminescence assay

MSCs were plated at $8.5 \times 10^4$ cells per well in 24-well plates (Corning). Lentivirus carrying the response elements for type I (ISRE) or type II (GAS) upstream of firefly luciferase was purchased from Qiagen. 24 hours post plating, cells were co-transduced with virus in the following groups: ISRE with scrambled shRNA, ISRE with *GRASLND* shRNA, GAS with scrambled shRNA, and GAS with *GRASLND* shRNA. 24 hours post-transduction, cells were rinsed once in PBS and fresh medium was exchanged. Three days later, medium was switched to expansion medium with 100 ng/mL IFN-β (PeproTech) for wells with ISRE or with 5 ng/mL IFN-γ (PeproTech) for wells with GAS. MSCs were cultured for another 22 hr, and then harvested for luminescence assay using Bright-Glo Luciferase Assay System (Promega). Luminescence signals were measured using the Cytation 5 Plate reader (BioTek).

### Western blot

On the day of harvest, cells were homogenized with complete lysis buffer in ice cold PBS: 10X RIPA buffer (Cell Signaling Technology), 100X phosphatase inhibitor cocktail A (Santa Cruz Biotechnology), and 100X Halt protease inhibitor cocktails (ThermoScientific). Lysates were subsequently centrifuged at 14,000 x g for 15 min at 4°C, and supernatants were collected and stored at −20°C until further processing. Western blot was serviced by RayBiotech with the following antibodies: primary anti-β-actin (RayBiotech), primary anti-RNF144A (Abcam), primary anti-PKR (alias anti-EIF2AK2) (RayBiotech) and secondary anti-rabbit-HRP (horse radish peroxidase) (RayBiotech).

### Statistical analyses

All statistical analyses were performed using R (*R Development Core Team, 2018*). Results from biochemical assays are depicted as mean ± SD. Results from qRT-PCR are depicted as fold-change with error bars calculated per Applied Biosystems manual instruction.

## Acknowledgements

We thank the Genome Technology Access Center at Washington University in St Louis, the Proteomics Core Laboratory, and the Hope Center Viral Vectors Core for their resources and support. The CRISPR-dCas9-VP64 system was a generous gift from Dr Charles Gersbach. We also wish to thank Sara Oswald for providing assistance in technical writing of the manuscript. This work was supported by the Arthritis Foundation, by NIH grants AR50245, AG15768, AG46927, AR072193, AR073752, and AR074992, and by the Nancy Taylor Foundation for Chronic Diseases.

## Additional information

### Funding

| Funder | Grant reference number | Author |
|--------|------------------------|--------|
| Arthritis Foundation | | Farshid Guilak |
| Nancy Taylor Foundation for Chronic Diseases | | Farshid Guilak |
| National Institutes of Health | AR50245 | Farshid Guilak |
| National Institutes of Health | AG15768 | Farshid Guilak |
| National Institutes of Health | AG46927 | Farshid Guilak |

| National Institutes of Health | AR072193 | Farshid Guilak |
| National Institutes of Health | AR073752 | Farshid Guilak |
| National Institutes of Health | AR074992 | Farshid Guilak |

The funders had no role in study design, data collection and interpretation, or the decision to submit the work for publication.

### Author contributions
Nguyen PT Huynh, Conceptualization, Formal analysis, Investigation, Visualization; Catherine C Gloss, Jeremiah Lorentz, Ruhang Tang, Formal analysis, Investigation; Jonathan M Brunger, Audrey McAlinden, Bo Zhang, Critical discussion; Farshid Guilak, Conceptualization, Supervision, Funding acquisition

### Author ORCIDs
Nguyen PT Huynh (iD) https://orcid.org/0000-0002-7254-1645
Farshid Guilak (iD) https://orcid.org/0000-0001-7380-0330

### Decision letter and Author response
Decision letter https://doi.org/10.7554/eLife.49558.sa1
Author response https://doi.org/10.7554/eLife.49558.sa2

## Additional files

### Supplementary files
• Supplementary file 1. RNA-seq databases on MSC chondrogenesis (GSE109503) (*Huynh et al., 2018b*) and SOX9 overexpression in ASCs (GSE69110) (*Ohba et al., 2015*).

• Supplementary file 2. Differentially expressed genes with *GRASLND* knockdown as compared to scrambled control. Data is expressed as log2FoldChange in KD compared to scrambled control (a positive log2FC means that the gene is upregulated in KD samples. A negative log2FC means that the gene is downregulated in KD samples).

• Transparent reporting form

### Data availability
Sequencing data have been deposited in GEO under accession codes GSE129985.

The following dataset was generated:

| Author(s) | Year | Dataset title | Dataset URL | Database and Identifier |
|---|---|---|---|---|
| Huynh NP, Gloss CC, Lorentz J, Tang R, Brunger JM, McAlinden A, Zhang B, Guilak F | 2020 | Long non-coding RNA GRASLND enhances chondrogenesis via suppression of interferon type II signaling pathway | https://www.ncbi.nlm.nih.gov/geo/query/acc.cgi?acc=GSE129985 | NCBI Gene Expression Omnibus, GSE129985 |

The following previously published datasets were used:

| Author(s) | Year | Dataset title | Dataset URL | Database and Identifier |
|---|---|---|---|---|
| Huynh NP, Zhang B, Guilak F | 2018 | High-Depth Transcriptomic Profiling Reveals the Temporal Gene Signature of Mesenchymal Stem Cells During Chondrogenesis | https://www.ncbi.nlm.nih.gov/geo/query/acc.cgi?acc=GSE109503 | NCBI Gene Expression Omnibus, GSE109503 |
| He X, Ohba S, Hojo H, McMahon AP | 2015 | Distinct regulatory programs for Sox9 in transcriptional regulation of the developing mammalian chondrocyte [RNA-seq] | https://www.ncbi.nlm.nih.gov/geo/query/acc.cgi?acc=GSE69110 | NCBI Gene Expression Omnibus, GSE69110 |

| Ramos YF, den Hollander W, Bovée JV, Bomer N, van der Breggen R, Lakenberg N, Keurentjes JC, Goeman JJ, Slagboom PE, Nelissen RG, Bos SD, Meulenbelt I | 2014 | Gene expression profiles from joint-matched macroscopically intact and OA affected cartilage of patients undergoing joint replacement surgery due to end-stage OA | https://www.ncbi.nlm.nih.gov/geo/query/acc.cgi?acc=GSE57218 | NCBI Gene Expression Omnibus, GSE57218 |
| --- | --- | --- | --- | --- |
| Dunn SL, Soul J, Anand S, Schwartz JM, Boot-Handford RP, Hardingham TE | 2017 | Gene expression changes in damaged osteoarthritic cartilage identify a signature of non-chondrogenic and mechanical responses | https://www.ebi.ac.uk/arrayexpress/experiments/E-MTAB-4304/ | ArrayExpress, E-MTAB-4304 |

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
