## [Decision Letter]

**Acceptance summary:**

Your work breaks new ground in understanding lncRNAs in bone and cartilage biology and provides a novel mechanism vis a vis interferon type II. Your overexpression of GRASLAND model that demonstrates augmentation of both mesenchymal stem cell and adipose derived stem cell chondrogenesis provides a promising approach to enhancing stem cell chondrogenesis and cartilage regeneration.

**Decision letter after peer review:**

Thank you for submitting your article "Long non-coding RNA GRASLND enhances chondrogenesis via suppression of interferon type II signaling pathway" for consideration by *eLife*. Your article has been reviewed by three peer reviewers, and the evaluation has been overseen by a Reviewing Editor and Harry Dietz as the Senior Editor. The following individuals involved in review of your submission have agreed to reveal their identity: James Dennis (Reviewer #2); Elizabeth G. Loboa (Reviewer #3).

The reviewers have discussed the reviews with one another and the Reviewing Editor has drafted this decision to help you prepare a revised submission.

The BRE and all three reviewers were enthusiastic about your manuscript because of its novelty and new insights. The experiments are well done and demonstrate that "GRASLND" regulates chondrogenesis in MSCs. Notwithstanding, each of the reviewers had comments that appear addressable within a relatively short period of time. But since the conclusion of the paper is that EIF2AK2 binding by GRASLND facilitates chondrogenesis by regulating the IFN pathway, it is essential that validation of an interaction b/w endogenous EIF2AK2 and endogenous GRASLND is occurring during the differentiation of the MSCs or ASCs into cartilage, we suggest an antibody to pull down endogenous EIF2AK2 followed by qRTPCR of the endogenous transcript that comes with it.

There were several other comments that warrant your attention:

Reviewer #1:

1) The authors switch between GRASLND and RN144-AS1 throughout the manuscript referring to their newly designated name, GRASLND, in the text of the Results section only to use RNF144-AS1 in the Figures. I would suggest using the original name and add the change to GRASLND in the Discussion.

A couple of sections of the paper are very vague. For example, "We successfully designed two target shRNAs for each of the three candidates, and one target for the other candidate (Figure 1—figure supplement 2). Please change the text stating the gene names, this simplifies what was done, the genes that were evaluated, and more importantly, helps the reader interpret the overall approach and findings.

Similarly, the first paragraph of the subsection “RNF144A-AS1 is crucial to and specifically upregulated in chondrogenesis”. Among those, two were downregulated and two were upregulated upon ectopic *Sox9* overexpression. Adding the gene names here removes ambiguity and helps the reader.

Reviewer #2:

There are a few items that could be clarified and some issues that this reviewer would like to see addressed.

1) I may have missed it somewhere, but it appeared that RP11-366L20.2 had a similar effect on GAG expression. Was there a particular reason this lncRNA was not examined in more detail?

2) One important issue not addressed in this study is what is the impact of GRASLND expression on actual chondrocytes. This is an important issue because chondrocytes are a likely target of therapeutic interventions.

3) To this reviewer, the in situ hybridization results (Figure 2) don't appear to match the results in Figure 1G where RNF144A-AS1 expression is much higher at day 7 than at day 1. The images in Figure 2A don't seem to match that. Is that because of the overall cell numbers in day 1 vs. 7?

4) From the IHC results on type II collagen expression in adipose-derived stem cells, it appears that the impact of GRASLND overexpression is actually quite minimal. The authors should bring that to readers attention since it means the effects might vary with other target cells, such as chondrocytes.

5) In the discussion of RNF144A-AS1 effects on chondrogenesis (subsection “RNF144A-AS1 is crucial to and specifically upregulated in chondrogenesis”, second paragraph) it seems that the authors have a chicken or the egg comment. Is RNF144A-AS1 affecting osteogenesis and adipogenesis or is RNF144A-AS1 being affected by the two differentiation processes?

6) Would direct inhibition of IFN gamma have the same chondrogenic effect on these cells?

Reviewer #3:

Minor comments to be addressed are with respect to further details for inclusion in the Materials and methods. In particular, information related to the number, ages, gender, and ethnicities of donors from whom MSC and ASC were obtained were not provided. Further details related to statistical analyses would also be helpful. For example, numbers of biological replicates and technical replicates were not obvious in the manuscript.

Other than these issues, the manuscript was well written and provides novel information to the field.

---

## [Author Response]

Reviewer #1:1) The authors switch between GRASLND and RN144-AS1 throughout the manuscript referring to their newly designated name, GRASLND, in the text of the Results section only to use RNF144-AS1 in the figures. I would suggest using the original name and add the change to GRASLND in the Discussion.

We appreciate that this inconsistency was pointed out. Given that our data show that RNF144A-AS1 does not regulate RNF144A expression, we feel that the term RNF144A-AS1 may not be completely accurate. Therefore, we have systematically changed all instances in the manuscript from RNF144A-AS1 to GRASLND. These changes have been marked in blue text throughout the manuscript.

A couple of sections of the paper are very vague. For example, "We successfully designed two target shRNAs for each of the three candidates, and one target for the other candidate (Figure 1—figure supplement 2). Please change the text stating the gene names, this simplifies what was done, the genes that were evaluated, and more importantly, helps the reader interpret the overall approach and findings.

Thank you for this suggestion. We have modified the text as suggested in the first paragraph of the subsection “GRASLND is crucial to and specifically upregulated in chondrogenesis”.

Similarly, the first paragraph of the subsection “RNF144A-AS1 is crucial to and specifically upregulated in chondrogenesis”. Among those, two were downregulated and two were upregulated upon ectopic Sox9 overexpression. Adding the gene names here removes ambiguity and helps the reader.

Thank you for this suggestion. We have modified the text as suggested in the first paragraph of the subsection “GRASLND is crucial to and specifically upregulated in chondrogenesis”.

Reviewer #2:There are a few items that could be clarified and some issues that this reviewer would like to see addressed.1) I may have missed it somewhere, but it appeared that RP11-366L20.2 had a similar effect on GAG expression. Was there a particular reason this lncRNA was not examined in more detail?

We chose not to examine this lncRNA in further detail in the present study as we could only validate our findings with one shRNA. We initially set the cutoff criteria to require that two different shRNA be validated in this regard. Furthermore, we found that this lncRNA was upregulated by *SOX9* overexpression, and we would have expected it to be downregulated if it is exhibiting a pattern of MSC markers in our data set. While this lncRNA appears to be an interesting candidate for future investigation, it remained outside the scope of our study, which was focused on potential targets with the highest probability of serving as regulators of MSC chondrogenesis.

2) One important issue not addressed in this study is what is the impact of GRASLND expression on actual chondrocytes. This is an important issue because chondrocytes are a likely target of therapeutic interventions.

In this study, we were focusing on the role of this lncRNA in MSC differentiation, which would likely involve different processes than those of primary chondrocytes. To address this question, we examined the expression of GRASLAND in primary human chondrocytes during monolayer-induced dedifferentiation or chondrocyte inflammation induced by IL-1. We did not find any significant differences in GRASLND expression in these cases.

**Author response image 1. respfig1:** No: Without IL-1. Yes: with IL-1. n=4 Early: P0 and P1. Late: P4 and P5. n=5.

3) To this reviewer, the in situ hybridization results (Figure 2) don't appear to match the results in Figure 1G where RNF144A-AS1 expression is much higher at day 7 than at day 1. The images in Figure 2A don't seem to match that. Is that because of the overall cell numbers in day 1 vs. 7?

Thank you for this comment. The presence of GRASLND is shown by the arrow, and the additional fluorescence is background. This signal was not observed on day 1.

Yes, the reviewer is correct regarding cell numbers. As is apparent by the Hoechst labeling, the signal is stronger at day 1 and day 7 compared to day 21, which is likely due to the higher cell density at earlier times. As chondrogenesis progresses, tissue accumulates and cells become more dispersed in the pellet. To maintain consistent readouts throughout the experiment, we used the same exposure for all three time points. For this reason, the background signal seems to be stronger in day 1 compared to day 7.

4) From the IHC results on type II collagen expression in adipose-derived stem cells, it appears that the impact of GRASLND overexpression is actually quite minimal. The authors should bring that to readers attention since it means the effects might vary with other target cells, such as chondrocytes.

We agree with the reviewer and appreciate this comment. While differences are not readily apparent by histology, we observed significant effects of GRASLND overexpression in increased levels of glycosaminoglycans as well as *COL2A1* expression, as measured quantitatively by biochemical assays and qPCR.

We have modified the text to reflect these changes in the subsection “GRASLND is crucial to and specifically upregulated in chondrogenesis”.

5) In the discussion of RNF144A-AS1 effects on chondrogenesis (subsection “RNF144A-AS1 is crucial to and specifically upregulated in chondrogenesis”, second paragraph) it seems that the authors have a chicken or the egg comment. Is RNF144A-AS1 affecting osteogenesis and adipogenesis or is RNF144A-AS1 being affected by the two differentiation processes?

This is an interesting point. In this study, we did not test whether osteogenesis or adipogenesis is regulated by GRASLND, but only that it is downregulated during this processes. To address this question directly, we would need to perform extensive additional experiments to knock down or overexpress GRASLND in these assays to prove it is having an effect on osteogenesis or adipogenesis, but we feel that these experiments are outside the scope of the present study.

6) Would direct inhibition of IFN gamma have the same chondrogenic effect on these cells?

This is an excellent suggestion. We did not perform these experiments but plan to examine this mechanism in future studies.

Reviewer #3:Minor comments to be addressed are with respect to further details for inclusion in the Materials and methods. In particular, information related to the number, ages, gender, and ethnicities of donors from whom MSC and ASC were obtained were not provided.

Thank you for this suggestion. We are unable to provide these data as the donors used in this study were completely de-identified, in compliance with HIPAA regulations and our IRB, so unfortunately this information was not available to us.

Further details related to statistical analyses would also be helpful. For example, numbers of biological replicates and technical replicates were not obvious in the manuscript.

We agree with the reviewer’s comment, and we have provided the number of biological replicates in the figure legends. For each of the biochemical or qRT-PCR assays, two technical replicates were performed for each biological replicates.